# Suitability Analysis of 17 Probiotic Type Strains of Lactic Acid Bacteria as Starter for Kimchi Fermentation

**DOI:** 10.3390/foods10061435

**Published:** 2021-06-21

**Authors:** Hee Seo, Jae-Han Bae, Gayun Kim, Seul-Ah Kim, Byung Hee Ryu, Nam Soo Han

**Affiliations:** 1Brain Korea 21 Center for Bio-Health Industry, Department of Food Science and Biotechnology, Cheongju 28644, Korea; gmldi873@naver.com (H.S.); wassup74@hanmail.net (J.-H.B.); lisugar@naver.com (G.K.); ksaha1004@naver.com (S.-A.K.); 2Fresh Food Research Division, Food BU, Daesang Corporation Research Institute, Icheon 17384, Korea; ryubh@daesang.com

**Keywords:** kimchi, starter, probiotics, health promotion, *Limosilactobacillus fermentum*, *Limosilactobacillus reuteri*

## Abstract

The use of probiotic starters can improve the sensory and health-promoting properties of fermented foods. This study aimed to evaluate the suitability of probiotic lactic acid bacteria (LAB) as a starter for kimchi fermentation. Seventeen probiotic type strains were tested for their growth rates, volatile aroma compounds, metabolites, and sensory characteristics of kimchi, and their characteristics were compared to those of *Leuconostoc* (*Le*.) *mesenteroides* DRC 1506, a commercial kimchi starter. Among the tested strains, *Limosilactobacillus fermentum, Limosilactobacillus reuteri, Lacticaseibacillus rhamnosus, Lacticaseibacillus paracasei,* and *Ligilactobacillus salivarius* exhibited high or moderate growth rates in simulated kimchi juice (SKJ) at 37 °C and 15 °C. When these five strains were inoculated in kimchi and metabolite profiles were analyzed during fermentation using GC/MS and ^1^H-NMR, data from the principal component analysis (PCA) showed that *L*. *fermentum* and *L*. *reuteri* were highly correlated with *Le*. *mesenteroides* in concentrations of sugar, mannitol, lactate, acetate, and total volatile compounds. Sensory test results also indicated that these three strains showed similar sensory preferences. In conclusion, *L. fermentum and L. reuteri* can be considered potential candidates as probiotic starters or cocultures to develop health-promoting kimchi products.

## 1. Introduction

Kimchi is a traditional Korean fermented vegetable made by fermenting salted cabbage, radish, and cucumber with various spices, including red pepper powder, garlic, ginger, and other ingredients [1]. Natural fermentation with unsterilized raw materials leads to the growth of various microorganisms, including lactic acid bacteria (LAB), resulting in inconsistent quality of kimchi, which may hamper the commercial production of high-quality products [2,3]. The use of starter cultures in the manufacture of kimchi has been considered an alternative to solve these problems [4]. Applying a starter culture to kimchi has several advantages, such as uniform quality, enhanced sensory characteristics, extended shelf-life, and functional properties of the kimchi product [5,6]. *Leuconostoc* spp. are the dominant species present in the early phase of kimchi fermentation at low temperatures; *Le. mesenteroides* and *Le. citreum* have been used as industrial starters since the early 2000s [7,8]. However, despite their excellent role as starters, they are not considered as health-promoting probiotics because of their absence of resistance to acid and bile salt and low colonization capacity in the large intestine [9]. Thereby, in order to manufacture health-promoting kimchi products, the development of a probiotic starter that can be used as an alternative of *Leuconostoc* spp. is required.

LAB are present in traditional fermented foods as major starter cultures and are commonly used as health-promoting probiotics. A report by the World Health Organization (WHO) defined probiotics as “live microorganisms which when administered in adequate amounts confer a health benefit on the host [10,11]”. Fermented foods can be used as delivery vehicles for probiotics [12,13]. A previous study reported that consumption of not only pure probiotics but also foods fermented using the same probiotic resulted in equal levels of functionality in piglets; administration of *Propionibacterium freudenreichii* in the form of bacteria cells and fermented cheese increased the population of Propionibacteria and the levels of total short-chain fatty acids in feces, and they provided anti-inflammatory activity [14]. Therefore, it is desirable to develop LAB that can be used as health-promoting starters and their application to produce functional fermented foods.

As for kimchi, the development of probiotic starters has been a leading interest among microbiologists to maintain product quality and to improve the health-promoting effects of kimchi [2]. For this purpose, researchers have developed various LAB as probiotic starters, such as *Bifidobacterium* spp. producing conjugated linoleic acid [15] and *Lactobacillus* spp. producing γ-aminobutyric acid (GABA) [16]. Additionally, for the same goal, mixed starter cultures have been employed to improve the sensory quality and anticancer and antioxidant activities of kimchi [17]. However, it has been difficult for the kimchi industries to find a successful commercialization case of probiotic starter to date. After investigation of these studies as well as communication with manufacturers, we concluded that those probiotic candidates providing health-promoting effects lack suitable starter properties essential for economical production in the industry. Briefly, a candidate strain of health benefit exhibited little or no growth in kimchi, and another probiotic resulted in a quality loss after fermentation because it produced off-flavor metabolites. Based on these findings, we propose the following requirements that the probiotic candidates should meet: the probiotic must (i) grow well in a kimchi environment containing various constituents, such as garlic and ginger; (ii) grow well at low temperature; (iii) prevent over-ripening by generating optimal concentrations of organic acids; (iv) provide favorable taste and aroma of kimchi; and (v) possess health-promoting properties. To develop suitable probiotic starters for kimchi fermentation, the above hypothesis should be assessed using well-established probiotic bacteria in kimchi.

The aim of this study was to evaluate the suitability of well-known probiotics as kimchi starters to produce health-promoting products with preferred taste. To this end, 17 probiotic type strains listed in the Health Functional Food Code in Korea (Table 1) were cultivated in simulated kimchi juice (SKJ) at 37 °C or 30 °C and 15 °C for 5 days, and their growth profiles, such as optical density and pH changes, were analyzed. Type strains exhibiting good growth in SKJ were used as starters for kimchi fermentation, and metabolite profiles were analyzed by GC/MS and ^1^H-NMR. Then, the effects of the type strains on sensory qualities were compared with kimchi fermented by a commercial starter, *Leuconostoc mesenteroides* DRC1506.

## 2. Materials and Methods

### 2.1. Microorganisms and Culture Conditions

The bacterial strains used in this study are listed in Table 1. The probiotic strains are known to provide health benefits when consumed, generally by improving or restoring the gut flora based on available information and scientific evidence [18,19,20]. They are approved as health functional food ingredients by Korean Ministry of Food and Drug Safety (MFDS). Probiotic type strains (17 species) were obtained from the Korean Agricultural Culture Collection (KACC, Wanju, Korea) or the Korean Collection for Type Cultures (KCTC, Daejeon, Korea). The commercial kimchi starter, *Leuconostoc mesenteroides* DRC1506, was kindly provided by Daesang FNF (Icheon, Korea). Most LAB were cultured in MRS broth (BD Difco, Detroit, MI, USA) at 37 °C for 24 h under anaerobic conditions. *Bifidobacterium* spp. were cultured in MRS broth (BD Difco) supplemented with 0.05% (*w*/*v*) L-cysteine hydrochloride (Sigma-Aldrich, St Louis, MO, USA).

### 2.2. Preparation of SKJ

SKJ was used as a pasteurized liquid medium simulating the kimchi environment for reproducible monitoring of microbial growth in the early stage of starter selection. SKJ was prepared according to the method described by Lee et al. [21]. Raw materials (cabbage, radish, garlic, ginger, and leek) and their contents are listed in Table 2. All materials were finely cut using a physical blender and incubated overnight after the addition of salts. Next, fish peptone (Bision, Seoul, Korea) was added instead of *jeotgal* (fermented fish source), and the mixture was pasteurized at 70 °C for 30 min. After cooling down to room temperature, the mixture was centrifuged at 7000× *g* for 10 min to remove the pulp, and the supernatant was used as the SKJ in the experiments. After pasteurization of the juice, no bacterial growth was confirmed by incubating SKJ on an MRS agar plate for two days.

Bacterial strains cultured in the media listed in Table 1 were washed twice with sterile phosphate-buffered saline (PBS, pH = 7.2), and 10^9^ colony forming units (CFU)/mL were inoculated in the SKJ. After cultivating these strains at their optimal temperatures, such as 37 or 30 °C as described in Table 1, optical density and pH were measured using a spectrophotometer (BioTek, Winooski, VT, USA) at 600 nm and a pH meter Orion Star A211 (Thermo Fisher Scientific Inc., MA, USA), respectively. In addition, to evaluate the adaptability of 17 probiotic LAB to low temperature, they were cultured at 15 °C for 5 days under the same condition, and optical density and pH were monitored periodically. In industry, kimchi is often fermented and distributed at 10 °C, where the growing bacteria are greatly limited. Therefore, in this study, they were cultured at 15 °C in SKJ as a preliminary step for the broad selection of probiotic kimchi starters that grow at low temperatures.

### 2.3. Preparation and Fermentation of Kimchi

Baechu kimchi was prepared according to the commercial kimchi manufacturing method of the Daesang FNF (Icheon, Korea) as described in Korean patent No. 10-1809447 [22]. Briefly, for the desalting process, baechu cabbages were washed with tap water and soaked in 8% salt solution at room temperature for 12 h. The salted cabbages were washed twice with tap water and then dehydrated. The salted cabbage (82.5%) was mixed with various condiments, such as pepper powder (2.5%), crushed garlic (2%), crushed ginger (0.5%), sliced green onion (2%), and sliced radish (9%). For kimchi fermentation, pre-cultured LAB were inoculated at a concentration of 10^5^ cells/g-kimchi and incubated at 10 °C. After reaching pH 4.4, an optimum condition for the ripened state, kimchi was stored at −0.5 °C for sensory testing, and aliquots were frozen at −80 °C for metabolite analyses. For pH measurement, kimchi samples (10 g) were diluted in 90 mL of 0.85% NaCl solution and were homogenized using the Seward Stomacher 400 Lab System (Seward, Norfolk, UK) for 3 min. The pH of the kimchi samples was measured using an Orion Star A211 pH meter (Thermo Fisher Scientific Inc. Waltham, MA, USA).

### 2.4. Analysis of Volatile Compounds by GC-MS

Volatile aroma compounds were extracted using solid-phase microextraction (SPME) fibers (DVB/CAR/PDMS, 50/30 μm, Supelco, 57298-U) and measured using a gas chromatography-mass spectrometer (GC/MS) (7820A/5977E MSD, Agilent Technologies, Palo Alto, CA, USA) with an Agilent GC Sampler 120 PAL autosampler. Before analysis, the injector was conditioned by exposure to 250 °C for 60 min. For the extraction of volatile compounds, kimchi juice (1 g) was placed in a headspace vial (20 mL, 22.5 mm × 75.5 mm) with 1 mL of distilled water and stirred at 300× *g* at 51 °C for 20 min. For quantitative analysis, 10 μL of 100 ppm methyl cinnamate (Sigma-Aldrich) dissolved in ethanol were added to distilled water as an internal standard. Then, the volatile components were adsorbed onto the SPME fiber for 30 min, and the fiber was automatically injected into the GC injector in the splitless mode for 2 min. For the GC/MS, a DB-WAX column (50 m × 200 μm × 0.2 μm, Agilent Technologies) was used with helium flowing at a rate of 1.5 mL/min. The injector temperature was 250 °C, and the oven temperature was increased from 40 °C (5 min) to 150 °C (0 min) at a rate of 5 °C/min followed by an increment to 200 °C (10 min) at a rate of 7 °C/min. The chromatograms of volatile compounds were acquired using a scan mode of *m*/*z* 33–250 at a fragment voltage of 70 eV, and peaks were identified through a library search (NIST ver. 11).

### 2.5. Metabolite Analysis by ^1^H-NMR

When kimchi reached the optimum ripening state (pH 4.4), non-volatile metabolite profiles, including organic acids, carbohydrates, and amino acids, were analyzed using ^1^H-NMR spectroscopy. Briefly, 3 mL of kimchi juice adjusted to pH 6.0 were mixed with 3 mL of deuterium oxide solution (99.9% D_2_O; Sigma-Aldrich) containing 1 mM sodium 2,2-dimethyl-2-silapentane-5-sulfonate (DSS; Sigma-Aldrich). The mixture was centrifuged for 5 min at 13,000× *g*, and 750 μL of the supernatant were transferred into a 5-mm NMR tube. Spectra were acquired using a Bruker 500-MHz NMR spectrometer (Bruker Magnetics, Faellanden, Switzerland) at 25 °C. Identification and quantification of individual metabolites from the ^1^H-NMR spectra were performed using the Profiler module of the Chenomx NMR Suite program (ver. 6.1; Chenomx, Edmonton, AB, Canada).

### 2.6. Sensory Evaluation of Kimchi

A total of 20 selected and semi-trained evaluators, the graduate students in the program of Food Sciences at Chungbuk National University, participated in the sensory evaluation of kimchi. They were trained on relevant guidelines prior to sensory evaluation. The protocol was approved by the Institutional Review Board of Chungbuk National University (IRB No. CBNU-202004-BMSB-0060-01). Optimally ripened kimchi (pH 4.4 ± 0.1) was used for testing, and the characteristics, such as sensory properties, the harmony of taste, and preferences of texture and flavor, were scored using a 9-point scale: 1 = very bad, 5 = moderate, and 9 = very good.

### 2.7. Statistical Analysis

To determine significant differences, statistical analysis was performed by one-way analysis of variance (ANOVA) with Duncan’s multiple range test (*p* < 0.05) using IBM SPSS software version 22 (SPSS Inc., Chicago, IL, USA). Principal component analysis (PCA) was performed using the prcomp command of the R statistical software (R Core Team, Vienna, Austria).

## 3. Results

### 3.1. Microbial Growth Rates in Simulated Kimchi Medium

To investigate the adaptability of 17 LAB in kimchi containing cabbage, radish, salt, and several condiments, the cell growth rates were measured in the SKJ at their optimal temperatures (30 or 37 °C) for 24 h (Figure 1). The commercial starter, *Le. mesenteroides* DRC 1506 was cultivated simultaneously as a reference strain. Among the tested strains, six, including DRC 1506, grew (OD_600_ > 0.4) after 24 h and their growth rates were in the following order: *L. reuteri*, *L. fermentum*, *L. rhamnosus*, *Le. mesenteroides*, *L. salivarius*, and *L. paracasei*. Along with their growth, pH values also decreased in a similar pattern. Among these strains, *L. reuteri* and *L. fermentum* showed superior growth rates compared with *Le. mesenteroides*, revealing their outstanding adaptability in nutrient environments in the SKJ. Meanwhile, the other strains, including four species of bifidobacteria, did not grow significantly in SKJ. Presumably, the medium compositions in SKJ may not have met the nutritional requirements of these strains.

Besides, the adaptability of 17 LAB at low temperatures was tested by cultivating in SKJ at 15 °C for 5 days (Figure 2). Among the tested strains, *Le. mesenteroides* showed the fastest growth rate to reach OD_600_ 0.6 after 4 days, showing its excellent adaptability to low temperature conditions. Likewise, three strains, *L. fermentum*, *L. rhamnosus*, and *L. paracasei*, grew at 15 °C in the SKJ. *Le. mesenteroides*-inoculated SKJ reached pH 4.4 after 1 day, which is the optimal acidity providing the preferred taste in kimchi [23]. Meanwhile, *L. plantarum*, *L. rhamnosus*, and *L. paracasei* took 2 days; *L. casei* took 4 days; and *L. fermentum* took 5 days (Figure 2b) to reach pH 4.4. Notably, *L. plantarum*, *L. casei*, *L. acidophilus*, and *L. bulgaricus* did not grow well at low temperature, but the pH decreased in the cultures. Compared to the other strains, *Le. mesenteroides* produced lactic acid more quickly. At the end of the fermentation at 15 °C, the pH of *L. rhamnosus* and *L. paracasei* was approximately 3.7 and that of *L. fermentum* was about 4.2, which did not show much difference from that of *Le. mesenteroides* (pH 3.7), indicating that an appropriate amount of acid was produced during fermentation. Therefore, *L. fermentum*, *L. rhamnosus*, and *L. paracasei* showed similar growth patterns to those of *Le. mesenteroides*.

### 3.2. Kimchi Fermentation by Inoculation of Five Strains

Based on the results of the above experiments, five strains capable of growing in SKJ at low temperature were selected as possible candidates as kimchi probiotic starters, namely, *L. fermentum*, *L. reuteri, L. rhamnosus, L. paracasei*, and *L. salivarius*. To investigate their effects on the fermentation profile and sensory changes in kimchi, these five strains and the commercial starter, *Le. mesenteroides*, were inoculated in cabbage kimchi at a concentration of 10^5^ cells/g, and the pH was monitored during kimchi fermentation at 10 °C (Figure 3). The initial pH of the kimchi samples was approximately 5.4 to 5.6, which did not change until the first day of fermentation, but from day 2 onwards, the pH of kimchi fermented with *Le. mesenteroides* and *L. fermentum* decreased rapidly and reached pH 4.4, on day 3. Meanwhile, the pH of kimchi fermented with *L. reuteri, L. rhamnosus, L. paracasei*, and *L. salivarius* decreased slowly and reached the optimal pH range (4.2~4.5) of kimchi on day 4.

### 3.3. Analysis of Metabolites in Kimchi

To evaluate the effect of starter on metabolite changes in kimchi, ^1^H-NMR analysis was conducted when they reached the optimal pH range (Table 3). The glucose content was low in both *Le. mesenteroides-* and *L. fermentum-*kimchi, revealing their rapid consumption rates for cell growth. Mannitol is a sugar alcohol produced by the reduction of fructose, which gives a fresh sweet taste in kimchi; its contents were high in *Le. mesenteroides-*, *L. fermentum-*, and *L. reuteri*-kimchi (28.96, 29.55, and 27.79 mM, respectively) compared with other kimchi. Lactate and acetate concentrations in the *Le. mesenteroides-*kimchi were 24.55 and 13.92 mM, respectively, showing a typical ratio in a hetero-lactic acid fermentation. Notably, lactate concentrations were significantly low in *L. rhamnosus* and *L. paracasei*-kimchi, probably affecting the quality of sour taste. Among the 15 amino acids detected, alanine, glutamate, glutamine, glycine, and serine were the most prevalent residues (>80%), and the glutamate content for umami taste was highest in *L. reuteri*-added kimchi.

A biplot of the principal component analysis (PCA) of metabolite compounds detected in kimchi is shown in Figure 4. Based on the PC1 axis, *Le. mesenteroides*- and *L. fermentum*-kimchi are located on the left, while the other kimchi are located on the right. Similarly, based on the PC2 axis, *Le. mesenteroides*- and *L. reuteri*-kimchi are located at the top, while the other kimchis are located at the bottom. In detail, *Le. mesenteroides* and *L. fermentum* were highly correlated with the levels of major metabolites, such as mannitol, ethanol, lactate, and acetate, and *Le. mesenteroides* and *L. reuteri* were highly correlated with glutamate and glutamine levels. In summary, *L. fermentum* and *L. reuteri*-kimchi share a close correlation with *Le. mesenteroides*-kimchi in terms of metabolite composition.

### 3.4. Analysis of Volatile Aroma Compounds

To examine the effect of starters on the aroma profiles of kimchi, the volatile compounds in the kimchi were analyzed by GC/MS when their pH reached the optimally ripened condition (Table 4). Seventeen compounds were detected, which were mostly sulfur compounds, such as dimethyl disulfide, diallyl sulfide, methyl 1-propenyl disulfide, methyl 2-propenyl disulfide, dimethyl trisulfide, diallyl disulfide, methyl 2-propenyl trisulfide, and methyl methylthiomethyl disulfide. The total amount of volatile compounds was lowest in *Le. mesenteroides*-kimchi (556.25 ± 14.68 ppm) and the levels were in the order of < *L. reuteri* (584.60 ± 32.49 ppm) < *L. rhamnosus* (657.38 ± 25.61 ppm) < *L. fermentum* (673.70 ± 36.52 ppm) < *L. paracasei* (875.84 ± 27.07 ppm) < *L. salivarius* (972.43 ± 18.71 ppm). In detail, the amounts of methyl 2-propenyl disulfide and diallyl disulfide were significantly lower in kimchi fermented with *Le. mesenteroides* and *L. reuteri*, and higher in kimchi fermented with *L. salivarius*. Notably, cyclopentyl isothiocyanate, which has a pungent odor, was low in kimchi fermented with *Le. mesenteroides, L. fermentum*, and *L. reuteri*, and high in kimchi fermented with *L. paracasei* and *L. salivarius*.

The profiles of all volatile compounds in the ripened kimchi samples were compared using PCA analysis. As shown in Figure 5, the kimchi fermented with *Le. mesenteroides, L. fermentum*, and *L. reuteri* were located on the top-left section of the biplot, while others were located on the right side where large amounts of sulfur compounds existed. The PCA biplot showed that *Le. mesenteroides, L. fermentum*, and *L. reuteri* had a high correlation with their volatile compound compositions, and relatively low amounts of sulfur compounds in the kimchi resulted in differences between kimchi samples.

### 3.5. Sensory Evaluation

A sensory test was conducted to evaluate the effect of different starters on kimchi taste (Figure 6). No differences were observed in the textural properties of cabbages between kimchi samples at optimal acidity, and the results indicated that the addition of starter did not significantly affect the physical properties of the kimchi (*p* < 0.05). However, as the result of the harmony of taste, *Le. mesenteroides, L. fermentum*, and *L. reuteri* were evaluated at 7.35, 6, and 5.45, respectively, and *L. rhamnosus*, *L. paracasei*, and *L. salivarius* were evaluated relatively low at 4.55, 4.25, and 4.2, respectively. Moreover, as a result of the overall preference evaluation, *Le. mesenteroides*, *L. fermentum*, and *L. reuteri* were evaluated at 7.6, 6.55, and 5.65, respectively, whereas *L. rhamnosus*, *L. paracasei*, and *L. salivarius* were scored relatively low at 4.5, 4.4, and 4.25, respectively. Overall, *Le*. *mesenteroides* was the most preferred kimchi starter, and *L*. *fermentum* and *L*. *reuteri* showed relatively high sensory properties. In particular, *L*. *fermentum* was not significantly different from that of *Le*. *mesenteroides*.

## 4. Discussions

Dairy products are usually fermented by primary starter LAB species, such as *Streptococcus thermophilus*, *Lactococcus lactis*, and *Lactobacillus delbrueckii* spp. *bulgaricus* [24,25]. These starter cultures are primarily used to produce lactic acid from lactose and may not necessarily possess probiotic properties [25]. In addition to the starter culture microorganisms, various probiotics, such as *Lacticaseibacillus casei*-group, can be added as adjunctive and/or secondary starters to improve the health functionality of fermented products [25,26]. Mixed culture fermentations provide complex growth patterns, which can also positively affect the organoleptic properties of the products [27]. Similarly, the application of carefully selected strains as functional starter cultures or co-cultures in kimchi fermentation processes will help to achieve the desired properties in situ, preserving a perfectly natural and healthy product. However, little research has been done on the effect of applying a mixed culture of commercial kimchi starter and probiotic strains on kimchi.

In this study, we evaluated the suitability of 17 probiotic type strains of LAB listed in the Health Functional Food Code in Korea as kimchi starters. Among the tested strains, *L. fermentum* KACC 11441 and *L. reuteri* KACC 11452 satisfied the four conditions that are necessary as a kimchi starter showing similar fermentation properties as those of *Le. mesenteroides* DRC 1506. In detail, the two species grew well in kimchi at low temperature, generated optimal concentrations of organic acids (lactate and acetate) by hetero-lactate fermentation, and provided the preferred taste and aroma of kimchi. Starter cultures used in kimchi need to adapt well to the unique environment of kimchi fermentation, which includes low temperature, low pH, and the presence of NaCl. Commercial kimchi products are typically distributed at low temperatures, and the organoleptic quality of kimchi fermented at low temperatures is superior to that of kimchi fermented at room temperature [2]. In addition, the taste of kimchi is mainly attributed to its metabolites, including sugars (glucose, fructose, and mannitol), organic acids (lactate and acetate), amino acids (glutamate), and CO_2_ (carbonate taste), and their contents are affected by the kimchi starter or microbial community during fermentation [3,6,28]. The main free sugars detected are glucose and fructose, which are important carbon sources metabolized by LAB during kimchi fermentation [3]. Notably, the high glucose content in *L. rhamnosus-*kimchi resulted in poor microbial growth during fermentation. Mannitol improves the taste of kimchi with a soft sweet taste that gives kimchi a refreshing sensation and suppresses the excessive sourness of kimchi [6,29]. This study demonstrates that *L. fermentum-* and *L. reuteri*-kimchi contained similar concentrations of glucose, mannitol, ethanol, lactate, acetate, and glutamate as those of the commercial kimchi fermented using *Le. mesenteroides* DRC 1506 starter. Moreover, sulfur compounds are the main volatile components of kimchi that are mainly derived from secondary ingredients, such as cabbage, garlic, and green onions. They play an important role in determining the quality of kimchi products because of their low threshold values and characteristic strong aroma [30]. Excessive amounts of sulfur compounds lower the sensory properties of kimchi, and some of the volatile compounds are regarded as off-flavors [31]. For instance, isothiocyanates are the major sulfur compounds responsible for the pungent flavor of cabbage, cauliflower, and collard greens, and they are produced by enzymatic hydrolysis of glucosinolates in these vegetables [32]. The optimal concentration of sulfur compounds with low levels of off-flavor chemicals is recommended for the preferred flavor of kimchi. Taken together, our results reveal that kimchi samples fermented by *L. fermentum* and *L. reuteri* have aroma compounds similar to those of commercial kimchi fermented by *Le. mesenteroides*. When the two species werecompared, *L fermentum* KACC 11441 showed the highest suitability as a kimchi starter because it grew faster at low temperature and showed similar total preferences with *L. mesenteroides* in sensory evaluation. Regarding the health-promoting effects of these species, additional analyses should be conducted, even though the Health Functional Food Code in Korea generally recognizes the health effects of strains belonging to the species of *L. fermentum* and *L. reuteri*.

*L*. *fermentum* and *L*. *reuteri* are Gram-positive, facultative anaerobic, and heterofermentative LAB. They are found in fermented vegetables, milk products, and human feces [33,34]. Recently, several studies have reported that *L. fermentum* is a good starter culture for traditional fermented vegetables in Vietnam and Africa [35,36]. *L*. *reuteri* has been used for the fermentation of pineapple, pumpkin, and oat-based juices, with no negative effects on the flavor [37,38,39]. Additionally, *L. fermentum* and *L. reuteri* strains have been proved to possess several health functionalities: they can tolerate acid and bile, adhere to the intestinal mucosa [40,41], lower cholesterol levels [42,43], ameliorate colonic inflammation [44,45], and prevent respiratory tract infections [46,47].

The limitation of this study is that type strains were used to test the suitability of LAB as kimchi starters. Type strains are descendants of the original isolates, which exhibit all the relevant phenotypic and genotypic properties listed in the original taxonomic circumscriptions published [48]. Recent studies have shown that the microbiological and biochemical characteristics differ strain specifically according to subspecies. For example, two *L. plantarum* subspecies isolated from kimchi had different characteristics [49], two *Lactococcus lactis* subspecies differed in response to stress [50], and two *B. longum* subspecies exhibited different inhibitory effects in a DSS-induced colitis mouse model [51]. Therefore, the results of this study only provide the general phenotypic and genotypic characteristics of the species tested, and for the development of superior starters for kimchi fermentation, further screening studies should be conducted against isolates belonging to *L*. *fermentum* and *L*. *reuteri*.

## 5. Conclusions

In this study, we evaluated the suitability of 17 probiotic type strains listed in the Health Functional Food Code in Korea for use as starters in kimchi. After five strains exhibiting good growth in SKJ were inoculated in kimchi, metabolite profiles and sensory qualities were compared to those of *Le. mesenteroides* DRC1506-kimchi. Our results show that *L. fermentum and L. reuteri* could be considered as possible candidates as kimchi starters. The two strains grew well in SKJ at low temperature and generated optimal concentrations of organic acids (lactate and acetate) providing the preferred taste and aroma of kimchi. *L. fermentum* and *L. reuteri* strains are known to have several health-related properties, such as acid-bile tolerance, intestinal adhesion, cholesterol-lowering ability, anti-inflammatory activity, and antimicrobial activity. To develop superior starters for kimchi fermentation to provide both good flavor and probiotic effects, further studies should be performed against isolates belonging to *L*. *fermentum* and *L*. *reuteri*.

## Figures and Tables

**Figure 1 foods-10-01435-f001:**
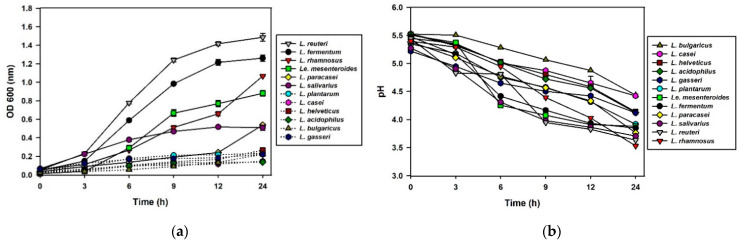
Cell growth (**a**) and pH changes (**b**) of the simulated kimchi juice (SKJ) after inoculation of 17 probiotic lactic acid bacteria (LAB) at their optimal growth temperatures, 37 or 30 °C. Optical density (OD_600nm_) values and pH values were measured for 24 h. The dotted lines represent little microbial growth (<OD 0.3) and pH changes (<pH 5.0). Results are expressed as means ± standard deviations (*n* = 3).

**Figure 2 foods-10-01435-f002:**
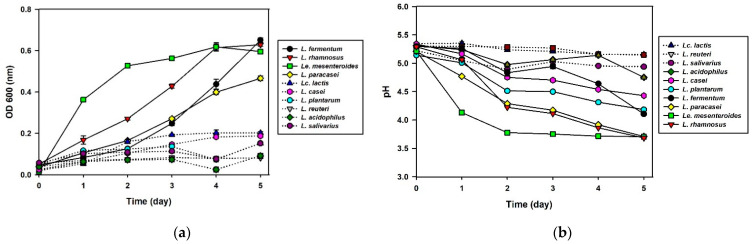
Cell growth (**a**) and pH changes (**b**) of the simulated kimchi juice (SKJ) after inoculation of 17 probiotic lactic acid bacteria (LAB) at low temperature, 15 °C. Optical density (OD_600nm_) values and pH values were measured for 24 h. The dotted lines represent little microbial growth (<OD 0.3) and pH changes (<pH 5.0). Results are expressed as means ± standard deviations (*n* = 3).

**Figure 3 foods-10-01435-f003:**
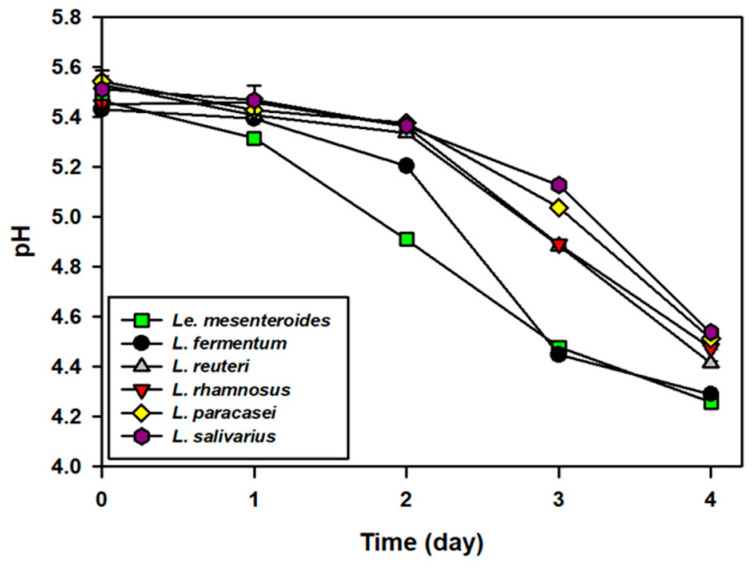
The pH changes in kimchi fermentation at 15 °C for four days after inoculation of five probiotic lactic acid bacteria (LAB). The LAB that showed fast growth rates in simulated kimchi juice (SKJ) (Figure 1) at low temperature (15 °C) (Figure 2) were *Limosilacotobacillus fermentum*, *Limosilacotobacillus reuteri*, *Lacticaseibacillus rhamnosus*, *Lacticaseibacillus paracasei* subsp. *paracasei*, and *Ligilactobacillus salivarius*. *Leuconostoc mesenteroides* DRC 1506, a commercial kimchi starter, was used for comparison. Results are expressed as means ± standard deviations (*n* = 3).

**Figure 4 foods-10-01435-f004:**
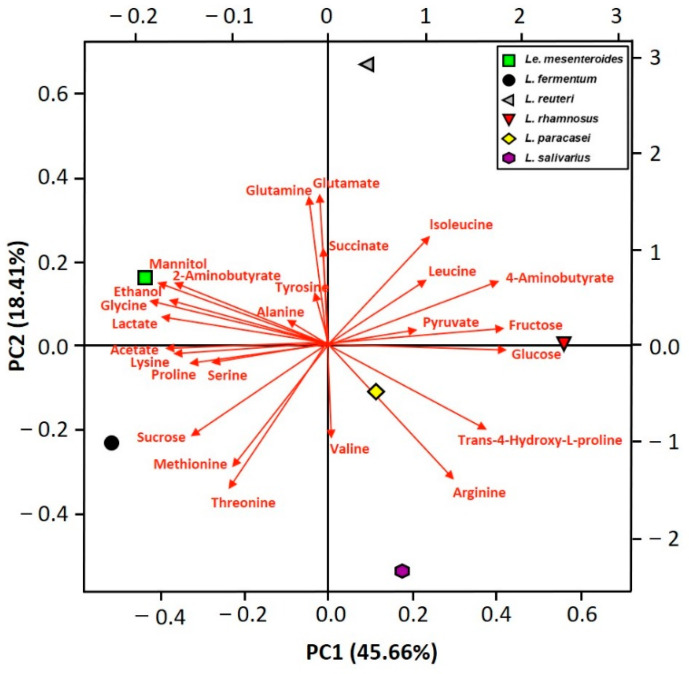
Biplot of the principal components analysis (PCA) of metabolite compounds detected in kimchi by ^1^H NMR after fermentation of five lactic acid bacteria (LAB) at 15 °C for four days. *Limosilacotobacillus fermentum*, *Limosilacotobacillus reuteri*, *Lacticaseibacillus rhamnosus*, *Lacticaseibacillus paracasei* subsp. *paracasei*, and *Ligilactobacillus salivarius* were tested, and *Leuconostoc mesenteroides* DRC 1506, a commercial kimchi starter, was used for comparison.

**Figure 5 foods-10-01435-f005:**
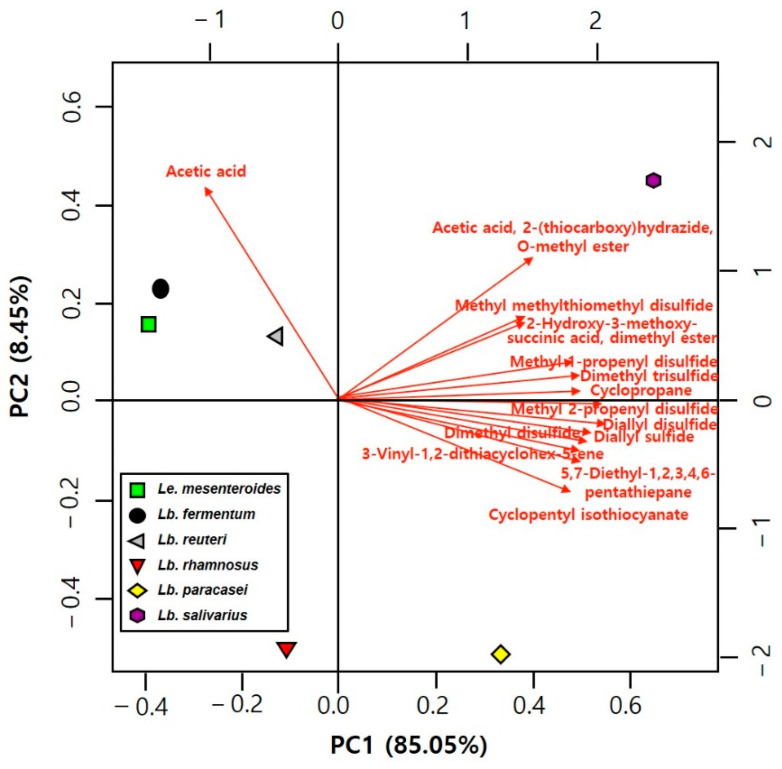
Biplot of the principal components analysis (PCA) on the volatile aroma compounds detected in baechu kimchi by GC/MS after fermentation of five lactic acid bacteria (LAB) at 15 °C for four days. *Limosilacotobacillus fermentum, Limosilacotobacillus reuteri, Lacticaseibacillus rhamnosus, Lacticaseibacillus paracasei* subsp. *paracasei*, and *Ligilactobacillus salivarius* were tested, and *Leuconostoc mesenteroides* DRC 1506, a commercial kimchi starter, was used for comparison.

**Figure 6 foods-10-01435-f006:**
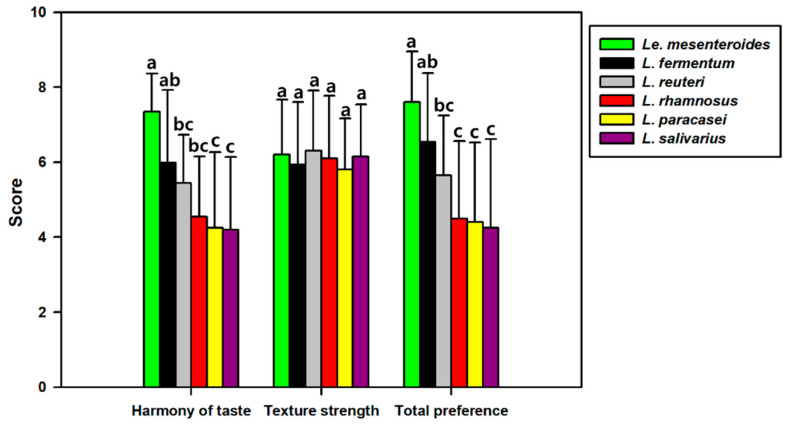
Sensory evaluation results of optimally ripened kimchi fermented by five lactic acid bacteria (LAB) at 15 °C for four days*. Limosilacotobacillus fermentum*, *Limosilacotobacillus reuteri*, *Lacticaseibacillus rhamnosus*, *Lacticaseibacillus paracasei* subsp. *paracasei*, and *Ligilactobacillus salivarius* were tested, and *Leuconostoc mesenteroides* DRC 1506, a commercial kimchi starter, was used for comparison. Results are expressed as means ± standard error of the mean (*n* = 20). Different letters indicate a significant difference at *p* < 0.05 according to Duncan’s multiple range test.

**Table 1 foods-10-01435-t001:** List of strains used in this study.

Species	Collections	Culture Condition (°C)
*Leuconostoc mesenteroides* DRC1506	KCCM11712P	30
*Limosilacotobacillus fermentum*	KACC11441	37
*Lactiplantibacillus plantarum*	KACC11451	37
*Limosilacotobacillus reuteri*	KACC11452	37
*Lacticaseibacillus rhamnosus*	KACC11953	37
*Lacticaseibacillus paracasei* subsp*. paracasei*	KACC12361	30
*Lacticaseibacillus casei*	KACC12413	30
*Lactobacillus helveticus*	KACC12418	37
*Lactobacillus acidophilus*	KACC12419	37
*Lactobacillus delbruechii* subsp*. bulgaricus*	KACC12420	37
*Lactobacillus gasseri*	KACC12424	37
*Ligilactobacillus salivarius*	KACC10006	37
*Lactococcus lactis*	KACC13877	30
*Streptococcus salivarius* subsp. *thermophilus*	KACC11857	37
*Bifidobacterium animalis* subsp. *lactis*	KACC16638	37
*Bifidobacterium breve*	KACC16639	37
*Bifidobacterium bifidum*	KACC20601	37
*Bifidobacterium longum*	KCTC3128	37

**Table 2 foods-10-01435-t002:** Composition of simulated kimchi juice (SKJ).

Ingredients	Concentration
Cabbage	700 g
Radish	200 g
Leek	50 g
Ginger	10 g
Garlic	20 g
Salt	3%
Fish peptone	0.5%

**Table 3 foods-10-01435-t003:** Quantitative values (mM) of metabolites in optimal ripened kimchi samples fermented with probiotic strains.

Groups	Metabolites	*Le. Mesenteroides*	*L. Fermentum*	*L. Reuteri*	*L. Rhamnosus*	*L. Paracasei*	*L. Salivarius*
Carbohydrates	Glucose	27.51 ± 1.99 ^c^	27.96 ± 2.06 ^c^	36.62 ± 0.39 ^b^	47.62 ± 7.26 ^a^	32.80 ± 1.99 ^bc^	38.61 ± 1.33 ^ab^
	Fructose	58.79 ± 1.22 ^b^	59.21 ± 0.40 ^b^	70.22 ± 1.23 ^a^	74.09 ± 5.62 ^a^	73.58 ± 2.76 ^a^	67.91 ± 0.61 ^a^
	Sucrose	4.28 ± 0.86 ^a^	4.77 ± 0.88 ^a^	4.14 ± 0.78 ^a^	4.01 ± 0.81 ^a^	4.45 ± 0.07 ^a^	4.34 ± 0.48 ^a^
Alcohols	Ethanol	16.62 ± 1.62 ^ab^	18.75 ± 0.13 ^a^	16.46 ± 1.39 ^ab^	14.17 ± 1.44 ^b^	14.24 ± 1.49 ^b^	14.21 ± 0.95 ^b^
	Mannitol	28.95 ± 2.22 ^a^	29.55 ± 2.33 ^a^	27.79 ± 1.20 ^a^	20.10 ± 0.83 ^b^	23.10 ± 1.32 ^b^	23.20 ± 0.59 ^b^
Organic acids	2-Aminobutyrate	1.80 ± 0.33 ^a^	1.68 ± 0.13 ^a^	1.64 ± 0.51 ^a^	1.26 ± 0.83 ^a^	1.71 ± 0.37 ^a^	1.38 ± 0.50 ^a^
	4-Aminobutyrate	1.93 ± 0.02 ^b^	1.62 ± 0.55 ^b^	4.15 ± 0.81 ^a^	4.15 ± 0.48 ^a^	2.85 ± 0.73 ^ab^	3.30 ± 0.60 ^a^
	Acetate	13.92 ± 1.64 ^ab^	16.63 ± 1.90 ^b^	12.86 ± 0.89 ^a^	10.44 ± 1.73 ^a^	10.34 ± 1.28 ^a^	12.24 ± 1.73 ^a^
	Lactate	24.55 ± 6.72 ^b^	21.92 ± 4.15 ^ab^	19.51 ± 1.24 ^ab^	16.16 ± 0.54 ^a^	16.32 ± 1.06 ^a^	18.81 ± 1.89 ^ab^
	Succinate	0.81 ± 0.24 ^a^	1.34 ± 0.43 ^a^	1.70 ± 0.10 ^a^	0.99 ± 0.38 ^a^	1.01 ± 0.51 ^a^	1.04 ± 0.58 ^a^
	Pyruvate	0.27 ± 0.09 ^a^	0.20 ± 0.08 ^a^	0.42 ± 0.11 ^a^	0.29 ± 0.14 ^a^	0.28 ± 0.02 ^a^	0.45 ± 0.25 ^a^
Amino acids	Alanine	7.36 ± 1.65 ^a^	8.72 ± 1.80 ^a^	8.56 ± 0.77 ^a^	7.71 ± 0.74 ^a^	7.89 ± 0.22 ^a^	7.91 ± 1.19 ^a^
	Arginine	1.86 ± 1.42 ^b^	2.43 ± 0.27 ^b^	1.88 ± 0.20 ^ab^	3.85 ± 0.29 ^a^	2.60 ± 0.51 ^ab^	3.90 ± 0.50 ^a^
	Glutamate	9.95 ± 3.12 ^ab^	11.49 ± 0.28 ^ab^	13.49 ± 0.81 ^a^	10.59 ± 0.73 ^ab^	10.04 ± 0.18 ^ab^	8.93 ± 1.91 ^b^
	Glutamine	13.69 ± 3.68 ^a^	9.42 ± 3.22 ^a^	12.85 ± 1.22 ^a^	11.60 ± 0.44 ^a^	10.04 ± 0.58 ^a^	9.50 ± 0.87 ^a^
	Glycine	5.74 ± 4.35 ^a^	5.37 ± 1.17 ^a^	4.34 ± 0.70 ^a^	1.70 ± 1.34 ^a^	4.03 ± 0.95 ^a^	3.01 ± 0.41 ^a^
	Isoleucine	0.63 ± 0.64 ^a^	0.67 ± 0.71 ^a^	1.07 ± 0.77 ^a^	0.91 ± 0.65 ^a^	0.59 ± 0.57 ^a^	0.77 ± 0.65 ^a^
	Leucine	0.83 ± 0.22 ^a^	0.80 ± 0.36 ^a^	1.70 ± 0.86 ^a^	1.06 ± 0.64 ^a^	1.20 ± 0.40 ^a^	1.40 ± 0.70 ^a^
	Lysine	1.70 ± 1.02 ^a^	1.16 ± 0.27 ^a^	0.75 ± 0.44 ^a^	0.64 ± 0.50 ^a^	0.92 ± 0.23 ^a^	0.92 ± 0.08 ^a^
	Methionine	0.62 ± 0.06 ^a^	1.20 ± 0.08 ^a^	0.56 ± 0.24 ^a^	0.31 ± 0.08 ^a^	0.99 ± 0.16 ^a^	1.09 ± 0.12 ^a^
	Proline	2.67 ± 1.08 ^a^	2.62 ± 0.75 ^a^	2.24 ± 0.33 ^a^	1.76 ± 1.07 ^a^	1.62 ± 0.68 ^a^	2.50 ± 0.76 ^a^
	Serine	15.70 ± 0.99 ^ab^	15.77 ± 0.79 ^ab^	15.82 ± 1.78 ^ab^	11.66 ± 0.95 ^b^	14.95 ± 2.03 ^ab^	16.50 ± 3.55 ^a^
	trans-4-Hydroxy-L-proline	0.31 ± 0.10 ^c^	1.08 ± 0.20 ^bc^	1.80 ± 1.18 ^abc^	2.94 ± 1.74 ^ab^	2.34 ± 0.59 ^ab^	3.61 ± 1.39 ^a^
	Threonine	2.74 ± 0.99 ^a^	3.94 ± 0.84 ^a^	1.46 ± 1.13 ^a^	1.72 ± 1.53 ^a^	2.05 ± 1.73 ^a^	3.89 ± 1.11 ^a^
	Tyrosine	0.45 ± 0.06 ^a^	0.46 ± 0.07 ^a^	0.42 ± 0.11 ^a^	0.47 ± 0.04 ^a^	0.55 ± 0.16 ^a^	0.29 ± 0.13 ^a^
	Valine	0.64 ± 0.43 ^a^	1.27 ± 0.30 ^a^	0.91 ± 0.47 ^a^	1.00 ± 0.63 ^a^	0.75 ± 0.44 ^a^	1.14 ± 0.53 ^a^

The different superscript letters indicate significant difference (*p* < 0.05) in the concentration of each metabolite.

**Table 4 foods-10-01435-t004:** Quantitative values (ppm) of volatile aroma compounds in optimally ripened kimchi samples fermented with probiotic strains.

Volatiles	*Le. Mesenteroides*	*L. Fermentum*	*L. Reuteri*	*L. Rhamnosus*	*L. Paracasei*	*L. Salivarius*
Cyclopropane	41.75 ± 2.71 ^b^	44.85 ± 1.97 ^b^	41.93 ± 0.90 ^b^	46.75 ± 2.84 ^b^	56.46 ± 3.52 ^a^	63.78 ± 2.53 ^a^
Propanamide, 2-hydroxy-	9.03 ± 0.68 ^b^	9.39 ± 2.17 ^b^	7.85 ± 0.83 ^b^	9.89 ± 0.08 ^b^	13.69 ± 0.54 ^a^	14.24 ± 0.75 ^a^
Dimethyl disulfide	26.55 ± 0.48 ^c^	36.31 ± 2.27 ^abc^	27.24 ± 1.73 ^c^	34.11 ± 2.18 ^bc^	44.42 ± 7.42 ^ab^	47.65 ± 0.83 ^a^
Diallyl sulfide	4.10 ± 0.37 ^c^	6.45 ± 0.49 ^bc^	4.86 ± 1.43 ^c^	6.63 ± 0.44 ^bc^	8.23 ± 1.27 ^ab^	9.58 ± 0.45 ^a^
Methyl 1-propenyl disulfide	3.65 ± 0.18 ^bc^	3.99 ± 0.17 ^b^	2.91 ± 0.13 ^c^	4.03 ± 0.55 ^b^	4.44 ± 0.07 ^b^	5.87 ± 0.31 ^a^
Methyl 2-propenyl disulfide	115.23 ± 5.53 ^c^	138.51 ± 6.52 ^b^	110.99 ± 8.88 ^c^	124.05 ± 4.50 ^bc^	166.52 ± 3.43 ^a^	176.20 ± 3.36 ^a^
Dimethyl trisulfide	54.83 ± 1.39 ^d^	74.08 ± 3.48 ^c^	59.35 ± 3.26 ^d^	64.89 ± 2.59 ^cd^	94.47 ± 3.85 ^b^	105.84 ± 3.19 ^a^
Acetic acid, 2-(thiocarboxy)hydrazide, O-methyl ester	13.93 ± 0.39 ^ab^	12.98 ± 0.72 ^ab^	14.26 ± 1.18 ^ab^	12.05 ± 0.63 ^a^	14.66 ± 0.27 ^b^	27.41 ± 0.84 ^a^
Acetic acid	6.12 ± 0.32 ^a^	5.86 ± 0.82 ^a^	6.72 ± 0.36 ^a^	3.68 ± 0.55 ^b^	3.34 ± 0.72 ^b^	5.18 ± 0.42 ^ab^
Diallyl disulphide	166.14 ± 3.53 ^c^	200.47 ± 14.13 ^b^	187.88 ± 12.63 ^bc^	213.83 ± 11.60 ^b^	281.32 ± 6.32 ^a^	307.74 ± 5.07 ^a^
2-Hydroxy-3-methoxy-succinic acid, dimethyl ester	9.10 ± 0.08 ^d^	10.82 ± 0.44 ^c^	10.52 ± 0.39 ^c^	9.23 ± 0.17 ^d^	12.76 ± 0.28 ^b^	14.85 ± 0.51 ^a^
Cyclopentyl isothiocyanate	5.55 ± 2.51 ^c^	7.57 ± 0.38 ^c^	7.81 ± 1.35 ^c^	12.39 ± 0.48 ^b^	16.90 ± 1.40 ^a^	18.01 ± 1.30 ^a^
Methyl 2-propenyl trisulfide	67.09 ± 2.19 ^c^	81.80 ± 4.46 ^b^	70.64 ± 4.71 ^bc^	77.56 ± 4.46 ^bc^	110.30 ± 5.19 ^a^	117.32 ± 3.02 ^a^
Methyl methylthiomethyl disulfide	2.43 ± 0.12 ^bc^	3.28 ± 0.22 ^b^	2.00 ± 0.13 ^c^	2.87 ± 0.22 ^bc^	2.86 ± 0.38 ^bc^	4.57 ± 0.35 ^a^
3-Vinyl-1,2-dithiacyclohex-5-ene	6.27 ± 1.77 ^b^	8.69 ± 0.58 ^ab^	6.45 ± 2.59 ^b^	10.68 ± 3.01 ^ab^	10.94 ± 0.78 ^ab^	13.69 ± 1.19 ^a^
1,2,3-Thiadiazole, 5-methyl-	11.10 ± 0.16 ^c^	12.55 ± 0.72 ^c^	11.49 ± 1.12 ^c^	12.35 ± 0.45 ^c^	15.81 ± 0.51 ^b^	18.49 ± 0.51 ^a^
5,7-Diethyl-1,2,3,4,6-pentathiepane	13.38 ± 0.73 ^c^	17.10 ± 0.75 ^b^	16.02 ± 1.47 ^b,c^	14.21 ± 0.48 ^c^	18.72 ± 0.60 ^b^	22.01 ± 0.40 ^a^
Total	556.25 ± 14.68 ^d^	673.70 ± 36.52 ^c^	584.60 ± 32.49 ^c,d^	657.38 ± 25.61 ^c^	875.84 ± 27.07 ^b^	972.43 ± 18.71 ^a^

The different superscript letters indicate significant difference (*p* < 0.05) in the concentration of each compound.

## Data Availability

The data in this study will be available upon request from the corresponding author.

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
