# Peer review of "Suitability Analysis of 17 Probiotic Type Strains of Lactic Acid Bacteria as Starter for Kimchi Fermentation"

_foods, 2021, doi:10.3390/foods10061435_

Round 1

Reviewer 1 Report

The manuscript: Suitability Analysis of 17 Probiotic Type Strains of Lactic Acid Bacteria as Starter for Kimchi Fermentation is well prepared and can be interesting to the readers.

Some major remarks:

  • are the strains used in the study probiotic? If yes, please give some information about properties of these strains and support it with sufficient references.
  • what is exactly new in your research? please highlight it in the introduction section
  • Table 1 and informations about strains should be moved to material and method section
  • In Methods section there is a lack of information about experiment in SKJ at 15^C for 5 days. But results are presented at Figure 2.
  • Results section should be clean from comments (some examples: lines 196-200; 209-210; 267-277, 310-321, etc....) . Please move these parts to discussion 
  • sensory analysis is poorly prepared and presented. Semi-consument test should incuded at least 40-50 participants. In my opinion sensory analisys shoul be ommited or changed.
  • lines 411-414 - please remove the sentence from conclusions becouse the statemant is not supported by the results

Some minor remarks:

  • line 46-47 please add references Hill et al., 2014
  • line 102 is pasterisation sufficient method to inactivate core microbiota?
  • why some experiments (in SKJ) were  set up at 15^C but the next one (kimchi fermentation) at 10^C?  

Reviewer 2 Report

Generally this paper is well written. I like the way they tested their hypothesis. The research topic is timely and the research question is interesting. There are not much research conducted with probiotic strains and their findings will add value to the scientific literature in the relevant field.

Introduction: Please provide the rational to sue these 17 probiotic type strains in this study very briefly.

Sensory study: Please indicate if this is trained panel or untrained panel ?

What is the purpose to use SKJ at the initial stage of this study? Please mention briefly.

Figures 1 & 2 seems bit crowded.

Reviewer 3 Report

The article presented by the authors is very interesting and is based on the evaluation of some LABs for application likes starter cultures in Kimichi production. Some points have to be improved.

-First of all the authors have to add an analysis of no volatile compounds that are very important for the technological, preservation, antioxidant properties of the product. I suggest to use an LC-MS-Q-TOF to perform that.

-Also the authors have to monitorize the presence pf the microorganism around all the shelf life of the product. It's a very important parameter to have a starter that could maintain important log growth around the shelf life of the products developed.

-I suggest also to test the antimicrobial properties of the strain used.

Round 2

Reviewer 1 Report

Thank you for improving the manuscript. In my opinion it is really good paper, however one minor comment should be addressed:

line 80- 93 The paragraph should be deleted. It contains informations included in Material and Methods section. Duplication of information should be avoided. The introduction part is good enough without that paragraph.

Reviewer 3 Report

The comments carried out by the authors are excellent.

Author Response

Thank you for your kind reply for our revised manuscript.